# Gauging the happiness benefit of US urban parks through Twitter

**Aaron J. Schwartz**[1,2,3,4,5]*, **Peter Sheridan Dodds**[3,4,6], **Jarlath P. M. O'Neil-Dunne**[2,3,5], **Taylor H. Ricketts**[2,5], **Christopher M. Danforth**[2,3,4,7]

**1** Ecology & Evolutionary Biology, University of Colorado, Boulder, Colorado, United States of America, **2** Gund Institute for Environment, University of Vermont, Burlington, Vermont, United States of America, **3** Vermont Complex Systems Center, University of Vermont, Burlington, Vermont, United States of America, **4** Computational Story Lab, MassMutual Center of Excellence for Complex Systems and Data Science, Vermont Advanced Computing Core, University of Vermont, Burlington, Vermont, United States of America, **5** Rubenstein School of Environment and Natural Resources, University of Vermont, Burlington, Vermont, United States of America, **6** Department of Computer Science, University of Vermont, Burlington, Vermont, United States of America, **7** Department of Mathematics & Statistics, University of Vermont, Burlington, Vermont, United States of America

* Aaron.J.Schwartz@colorado.edu

**Data Availability Statement:** The data relevant to this study are available from the Figshare repository at https://doi.org/10.6084/m9.figshare.

## Abstract

The relationship between nature contact and mental well-being has received increasing attention in recent years. While a body of evidence has accumulated demonstrating a positive relationship between time in nature and mental well-being, there have been few studies comparing this relationship in different locations over long periods of time. In this study, we analyze over 1.5 million tweets to estimate a happiness benefit, the difference in expressed happiness between in- and out-of-park tweets, for the 25 largest cities in the US by population. People write happier words during park visits when compared with non-park user tweets collected around the same time. While the words people write are happier in parks on average and in most cities, we find considerable variation across cities. Tweets are happier in parks at all times of the day, week, and year, not just during the weekend or summer vacation. Across all cities, we find that the happiness benefit is highest in parks larger than 100 acres. Overall, our study suggests the happiness benefit associated with park visitation is on par with US holidays such as Thanksgiving and New Year's Day.

## Introduction

The global COVID-19 pandemic has emphasized the importance of outdoor spaces for our collective well-being. With most people living in urban areas, parks have become the primary form of accessible nature for physical, recreational, social, and cultural activities. Urban greenspaces can provide opportunities to reduce the impacts of the "urban health penalty," the higher levels of stress and depression in city dwellers [1]. The benefits of nature contact occur within a specific geographic, cultural, and environmental context; however, most studies to date have focused on individual cities.

12915329.v2 (https://figshare.com/articles/dataset/
Tweet_Message_IDs/12915329).

**Funding:** AJS is supported by NSF GRFP program
DGE1451866. https://www.nsfgrfp.org/ All authors
are supported by Gund Institute for the
Environment Catalyst Award program. https://
www.uvm.edu/gund/gund-catalyst-awards The
funders had no role in study design, data collection
and analysis, decision to publish, or preparation of
the manuscript.

**Competing interests:** Authors associated with the
MassMutual Center for Excellence for Complex
Systems and Data Science are supported in part by
a gift from MassMutual Life Insurance Company.
This does not alter our adherence to PLOS ONE
policies on sharing data and materials.

Researchers have used experimental, epidemiological, and experience-based approaches to build a consensus around the mental health benefits of urban nature [2, 3]. Experimental studies are well-suited for measuring physiological and psychological responses to discrete periods of nature contact. Field experiments randomly assign participants to a nature-treatment group and an urban control group. Investigators then measure health or cognitive markers pre- and post-exposure [4]. Experimental designs usually examine the effects of nature contact following, rather than during exposure, though benefits experienced during exposure may be of interest [5]. Nevertheless, experiments have built a strong evidence base for the positive short-term mental health benefits from nature contact [3].

Epidemiological studies model the relationships between nature contact and health using surveys and geographic data. Compared to experiments, these studies estimate the health effects of nature contact over a longer time period with larger sample sizes. By measuring vegetative cover near where people live, these types of studies can utilize a continuous metric of nature contact. However, the existence of nearby vegetation does not guarantee its use, and most of these studies have focused on vegetation rather than access to nearby parks [6, 7]. Across four cities, researchers found varying effect sizes for the associations between nearby nature measured by vegetation and bird biodiversity and well-being [8]. A review found that while local-area greenspace was positively associated with mental well-being across multiple studies, the evidence is not currently sufficient enough to guide planning decisions [9].

Researchers have developed mobile applications to capture real-time emotional state in different locations. These methods, known as experience-based approaches, can collect high-resolution individual data in a variety of places and pair location tagged well-being data with demographic surveys conducted on participants [3]. The *Mappiness* and *Urban Mind* studies both found participants to be happier in natural environments, compared to urban [10, 11]. By observing participants longitudinally, these studies have the potential to provide insights into the underlying psychological mechanisms around happiness, location, and behavior [12].

Publicly available data from social media have been used to study human behavior in a variety of contexts, and have the potential to augment our understanding of the benefits of nature contact. Data from social media, and specifically Twitter, have been used to estimate visitation rates and activity types in urban greenspaces [13–15]. Several studies have begun to analyze tweet text to understand emotions in urban greenspaces. In Melbourne, tweets in greenspaces had higher positive emotions compared to tweets in urban areas [16]. In London, tweets from parks more frequently exhibited the positive emotions of surprise, joy, and anticipation [17]. Our prior work showed that in-park tweets in San Francisco were happier than tweets before and after park visits at other locations [18].

However, the relative mental benefits of nature contact across a wide geographic range have not been fully explored. The ability to access and enjoy nature is heterogeneous across cities—urban park systems vary widely in quality and investment [19]. Further study with social media data, which are available at a wide geographic and temporal scale, can complement experimental, epidemiological, and experience-based studies. In the present study, we examine the mental benefits of visiting urban parks using Twitter across 25 cities. Specifically, we pose 4 hypotheses:

**H1: We hypothesize that in-park happiness will be higher than out-of-park happiness across all cities**. We investigate whether prior results finding an association between nature contact and happiness holds across a wide geographic range. We next explore factors that may influence the relative differences in mental benefits across cities.

**H2A: We hypothesize that cities with higher levels of investment in parks will provide greater benefits to the mental well-being of park visitors**. A recent study found that

county area park expenditures were associated with better self-rated health [20]. Understanding inter-city variation in the mental health benefits of nature contact can inform urban planning and public health policy.

**H2B: We hypothesize that cities with higher levels of park quality will provide greater mental benefits for park visitors**. Other studies have suggested that parks with greater amenities and access will provide residents with greater opportunities for nature contact.

**H3: Across 25 cities, we hypothesize that larger parks will provide greater mental benefits than smaller parks**. Experimental approaches to nature contact are limited in the number of natural areas they can integrate into their study designs, while epidemiological studies rely on nearby nature and do not detect visits to specific locations [21]. Parks vary significantly in their size, amenities, and vegetative cover [22]. While it is difficult to capture all of these factors across many park systems, size can be a good proxy for the type and general function of a park. In our prior study, we found that the visitors to the largest official group of parks, known as Regional Parks, exhibited the greatest mental benefits in San Francisco [18]. However, it is again unclear whether this pattern will hold across other cities.

**H4: We hypothesize the mental benefits of park visitation to be the highest on the weekends and during the summer, but positive at all times**. Studies using data from mobile phone applications and Twitter have sampled over a time period between weeks and months and have not verified whether the timing (e.g., hour of day, day of week, time of year) of park visits impacts potential health benefits. However, a study using tweets in Melbourne demonstrated heterogeneity in emotional responses to nature across different seasons and time of day [16]. In addition, comparing the benefits of park visitation temporally is a way to check the extent to which observed happiness in parks is a function of park visits occurring during the weekend or summer vacation.

Here, we expand our prior work in San Francisco to the 25 largest cities in the US by population and compare tweets over a 4 year period [18]. For each city, we estimate a similar metric of *happiness benefit* to test the 4 hypotheses above.

## Materials and methods

### Data collection & processing

We used a database of tweets collected from January 1 2012 to April 27 2015 (Appendix A in S1 Appendix), limiting our search to English language tweets that included GPS coordinate location data (latitude and longitude). We chose this time period because geo-located tweets became abundant nationally in 2012 and dropped significantly in April 2015 when Twitter made precise location sharing an opt-in feature. Using boundaries from the US Census, we collected tweets within each of the 25 largest cities in the US by population [23]. We did not include retweets (tweets that are re-posted from another user) in our analysis.

We detected whether a tweet was posted within park boundaries using the Trust for Public Land's Park Serve database. Our ability to find tweets posted from inside parks depends on the accuracy of mobile GPS hardware which can vary by manufacturer, surrounding building height, and weather conditions. While most message locations should be precise to within 10m, some of our user pool may have posted just outside of parks due to measurement error. Data analysis of hashtag frequency revealed that a large number of geo-located tweets were posted by automated accounts (or bots) posting about job opportunities and traffic; any tweet found with a job or traffic related hashtag was removed from the sample (Appendix C in S1 Appendix).

**Table 1. Summary of geolocated Twitter data for the 25 most populous cities in the U.S. from 2012–2015.** 'Total tweets' enumerates all public tweets posted from a GPS latitude/longitude inside that city. 'Park tweets' is the total number of tweets posted from inside parks. The '% tweets in park' column calculates Park tweets / total Tweets. 'Park visitors' is the number of unique users who tweeted inside one of that city's municipal park locations as defined by Trust for Public Land's ParkServe. 'Parks visited' is the number of unique facilities from which a tweet was posted within that city. 'Tweets per capita' is number of total messages for the entire period divided by the city's population in 2012.

| City | Total tweets | Park tweets | % tweets in parks | Park visitors | Parks visited | Tweets per capita |
|---|---|---|---|---|---|---|
| New York | 2,892,512 | 213,813 | 7.4 | 113,702 | 1,880 | 0.35 |
| Los Angeles | 1,215,288 | 53,988 | 4.4 | 36,271 | 540 | 0.32 |
| Philadelphia | 1,166,125 | 64,857 | 5.6 | 26,287 | 482 | 0.76 |
| Chicago | 1,130,611 | 66,100 | 5.8 | 36,919 | 872 | 0.41 |
| Houston | 821,433 | 39,581 | 4.8 | 13,464 | 501 | 0.38 |
| San Antonio | 589,595 | 23,566 | 4.0 | 12,763 | 268 | 0.43 |
| Washington | 570,157 | 74,937 | 13.1 | 41,062 | 370 | 0.92 |
| Boston | 547,625 | 52,689 | 9.6 | 23,479 | 682 | 0.87 |
| San Diego | 491,219 | 36,080 | 7.3 | 22,269 | 406 | 0.37 |
| Dallas | 490,918 | 21,787 | 4.4 | 12,211 | 346 | 0.40 |
| San Francisco | 486,782 | 59,412 | 12.2 | 36,175 | 407 | 0.59 |
| Austin | 449,853 | 23,547 | 5.2 | 14,689 | 289 | 0.55 |
| Baltimore | 333,734 | 12,965 | 3.9 | 5,135 | 260 | 0.53 |
| Fort Worth | 320,178 | 9,664 | 3.0 | 4,278 | 239 | 0.42 |
| Phoenix | 268,455 | 12,041 | 4.5 | 7,566 | 189 | 0.18 |
| Columbus | 251,573 | 8,884 | 3.5 | 4,340 | 328 | 0.31 |
| San Jose | 234,234 | 8,263 | 3.5 | 4,517 | 314 | 0.24 |
| Indianapolis | 225,931 | 11,560 | 5.1 | 5,660 | 183 | 0.27 |
| Charlotte | 218,310 | 8,039 | 3.7 | 3,868 | 190 | 0.29 |
| Seattle | 201,533 | 12,758 | 6.3 | 7,739 | 373 | 0.32 |
| Detroit | 195,572 | 7,885 | 4.0 | 3,819 | 234 | 0.28 |
| Jacksonville | 194,777 | 6,219 | 3.2 | 3,218 | 261 | 0.23 |
| Memphis | 137,222 | 5,614 | 4.1 | 3,112 | 163 | 0.21 |
| Denver | 131,240 | 6,243 | 4.8 | 3,902 | 279 | 0.21 |
| El Paso | 96,015 | 2,722 | 2.8 | 1,397 | 180 | 0.14 |

We assigned a control tweet to each in-park tweet. For each tweet, we chose the closest-in-time out-of-park tweet from another user, temporally proximate to the in-park tweet within the same city. This message functions as a control because it allows us to compare the happiness of our in-park sample with a set of tweets that were posted in the same city and at roughly the same time. We summarize each city's Twitter data in Table 1. In Appendix D in S1 Appendix, we describe an alternative control group specification that uses out-of-park tweets from the same users who posted tweets inside of parks.

## Sentiment analysis

To approximate the mental benefits of park visitation, we used sentiment analysis, a natural language processing technique that associates numerical values with the emotional value of individual words. For the present study, we used the Language Assessment by Mechanical Turk (labMT) dictionary. LabMT includes happiness ratings of the most 10,222 commonly used English words. Words were rated independently by 50 people using Amazon's Mechanical Turk service on a scale of 1 (least happy) to 9 (most happy) [24]. For example, *beautiful* has an average happiness score of 7.92, *city* has an average happiness score of 5.76, and *garbage* has an average happiness score of 3.18 in labMT. We excluded words with scores between 4.0 and

6.0 from our analysis because they are emotionally neutral or particularly context dependent. The labMT sentiment dictionary performs well when compared with other sentiment dictionaries on large-scale texts, and correlates with traditional surveys of well-being including Gallup's well-being index [25, 26]. Sentiment analyses can be sensitive to small word sample sizes; therefore we apply labMT to collections of many tweets at once rather than individual tweets.

For each round of analysis, we aggregated tweets into an in-park group and a control group. We calculated the average happiness for each group of tweets as the weighted average of their labMT word scores using relative word frequencies as weights:

$$h_{\mathrm{avg}} = \frac{\sum_i^N h_i \cdot f_i}{\sum_i^N f_i}, \tag{1}$$

where $h_i$ is the happiness score of the ith word and $f_i$ is its frequency in a group of tweets with $N$ words. Next, we subtracted the average happiness of the control tweets from the average happiness of the in-park tweets and defined this difference as the "happiness benefit". To estimate uncertainty in our calculation of happiness benefit, we applied a bootstrapping procedure: We randomly sampled 80% of tweets without replacement from a set of in-park tweets and their respective control tweets and then re-calculated the happiness benefit. Performing this procedure 10 times, we derived a range of plausible happiness benefit values. Robustness checks were performed to show the convergence of this range at 10 runs.

We used the above technique to calculate the happiness benefit for all cities together and each city individually. For each city, we removed all words appearing in that city's park name before estimating the happiness benefit. For example, we removed *golden*, with an average happiness of 7.3, from all San Francisco tweets because of Golden Gate Park. The word *park* is also removed from all tweets. We performed a manual check on the top ten most influential words in a city's happiness benefit calculation. This allowed us to identify potential biases introduced by words being used in an unexpected manner. For example, we removed *ma* from all Boston tweets because it appears with a high frequency as an abbreviation for Massachusetts, but has a positive happiness score as shorthand for *mother*. We describe our methods and include the full list of stop words in Appendix B and Table 1 in S1 Appendix.

## Park analysis

We used data from the Trust for Public Land (TPL) to further investigate the happiness benefit from urban park visits. The TPL provides a variety of data on municipal park systems. Annually, TPL publishes a ParkScore® for the largest cities in the US, which is a composite score out of 100 that combines metrics of park size, access, investment, and amenities. We conducted a correlation analysis for city-level happiness benefit against 2018 ParkScore® and park spending per capita, also sourced from the TPL [27]. ParkScore® and spending for Indianapolis was sourced from TPL's 2017 data release due to lack of participation in 2018.

To investigate the relationship between happiness benefit and park size, we assigned every in-park tweet a category based on the size of the park from where it was posted. We grouped parks into four categories ($< 1$ acre, between 1 and 10 acres, between 10 and 100 acres, and greater than 100 acres). To have roughly equal representation from each city, we randomly selected tweets (along with their control tweet) in each park category from each city (or all of the tweets in that category if there were less than 500). After combining the randomly selected tweets from each city for each park category, we estimated the happiness benefit using the same bootstrapping procedure described above.

## Temporal analysis

Next, we estimate the happiness benefit based on when tweets were posted in three different ways. First, we grouped tweets based on the month they were posted in four seasonal groups (Winter: Dec, Jan, Feb; Spring: Mar, Apr, May; Summer: Jun, Jul, Aug; Fall: Sep, Oct, Nov). Second, we grouped tweets based on the day of the week they were posted. Finally, we grouped tweets based on the hour of the day they were posted in their local timezone (Appendix E in S1 Appendix). To have roughly equal representation from each city, we randomly selected 1,000 tweets (along with their control tweet) in each time category from each city (or all of the tweets in that category if there were less than 1,000). After combining the randomly selected tweets from each city, we estimated the happiness benefit using the same bootstrapping procedure described above.

## Results and discussion

In this study, across the 25 largest cities in the US, we find that people write happier words on Twitter in parks than they do outside of parks. This effect is strongest for the largest parks by area—greater than 100 acres. The effect is present during all seasons and days of the week, but is most prominent during the summer and on weekend days.

### Sentiment analysis

Across all cities, the mean happiness benefit was 0.10 (Bootstrap Range [.098, .103]), supporting **H1**. Across our 25 city sample, the mean happiness benefit ranged from 0.00 to 0.18. Indianapolis had the highest mean happiness benefit, while Baltimore had the lowest (Fig 1). Cities with more in-park tweets to sample from had tighter happiness benefit ranges, as exhibited by Denver, New York, Los Angeles, and Philadelphia. The mean happiness benefit was positive across all cities.

Pooling tweets across cities, we find a mean happiness benefit of 0.10. According to Hedonometer.org, which tracks Twitter happiness as a whole using the labMT dictionary, Twitter has fluctuated around a mean happiness of 6.02 since 2008. New Year's Day has historically had an average happiness of 6.11, giving it an average happiness benefit of .10. Christmas, historically the happiest day of the year on Twitter, has had an average happiness benefit of 0.24. The global COVID-19 Pandemic gained rapid recognition in the US on March 12, 2020, which resulted in the then unhappiest day in Twitter's history with a drop of 0.31 from its historical average. Following the murder of George Floyd, the Black Lives Matter protests led to a new all-time low, 0.39 below the historical average [28]. These are considered large swings, and we assert that the happiness benefit of 0.10 across a sample of 25,000 tweets is a strong signal. Prior work has shown that tweet happiness can vary within a city (even down to the neighborhood level) and extreme sentiment values may be obscured by our weighted averaging procedure [29]. We chose an aggregated approach to detect an overall signal about the effect of parks on happiness, but understanding detailed spatial patterns in happiness is an important future research direction. Positive words such as *beautiful, fun*, and *enjoying* contributed to the higher levels of happiness from our in-park tweet group. These words may relate to the stimulating aspects of urban greenspace. This is supported by a recent study that analyzed tweets to investigate which aspects of restoration were most prominent in urban greenspace. They found that fascination, an emotional state induced through inherently interesting stimuli, was most salient [30]. Fascination is one characteristic of nature experiences described by Attention Restoration Theory, which theorizes that time in nature provides an opportunity to recover from the cognitive fatigue induced by mentally taxing urban environments [31, 32].

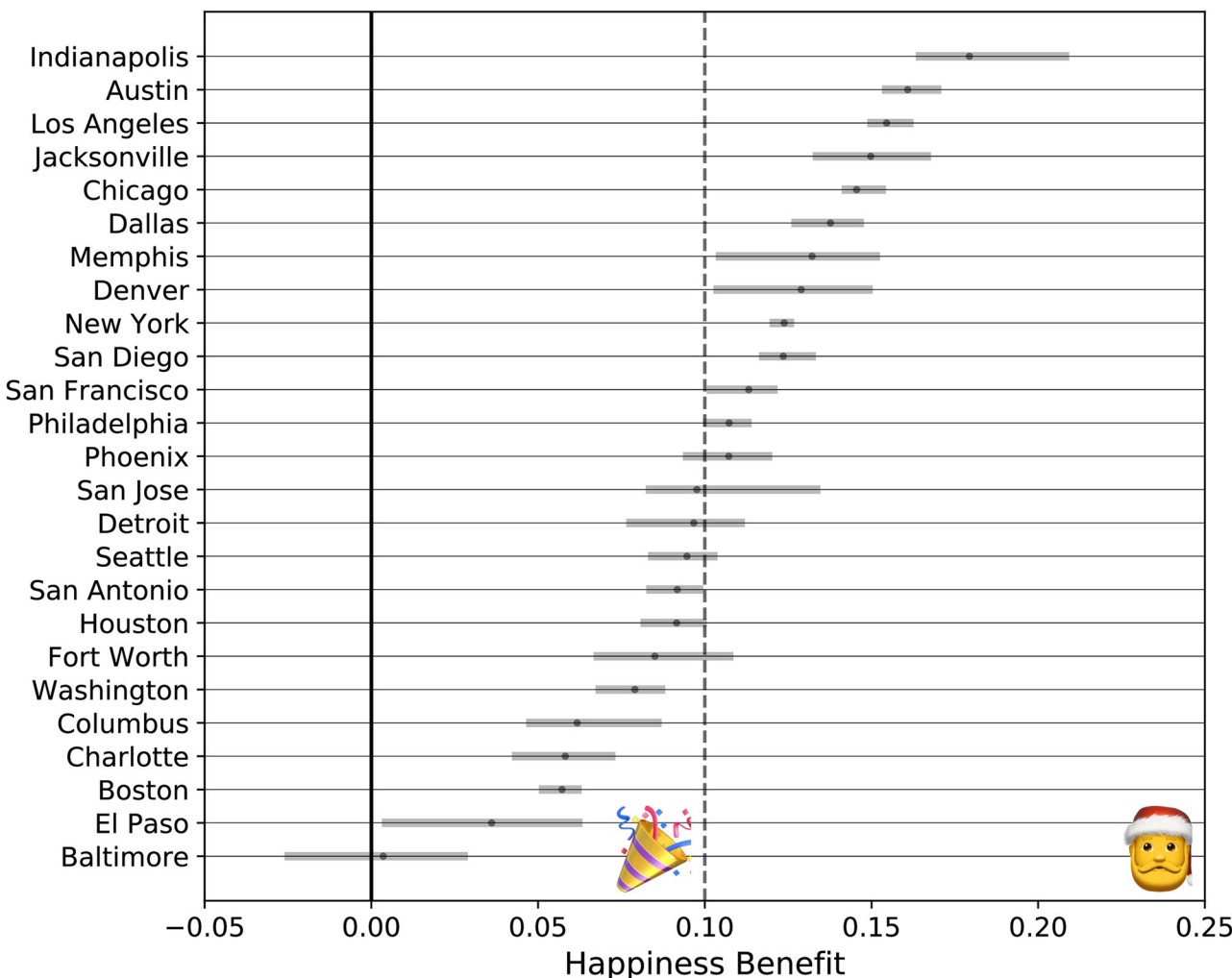

**Fig 1. Happiness benefit by city.** Happiness benefit is the difference in happiness score between in-park tweets and out-of-park tweets sent at roughy the same time. The full range of values is estimated from 10 bootstrap runs in which 80% of tweets were randomly selected. The dark grey dots represent the mean value from bootstrap runs. The solid line marks a happiness benefit of 0, and the dotted line is average happiness benefit across all 25 cities. Emojis denote the happiness benefit typically observed on New Year's Day and Christmas for all English tweets.

We find high levels of variation across cities for the happiness benefit between in-park and out-of-park tweets. For example, Chicago had higher frequencies of words such as *beautiful* driving higher in-park tweet happiness. Park tweets had lower frequencies of negative words such as *don't, not*, and *hate*. Psychological experiments treat positive and negative affect as separate measures [33]; the heterogeneity of the words driving the happiness benefit may be related to how these components of affect are being expressed via tweets.

## Wordshifts

The happiness benefit is driven by word frequency differences between the in-park tweets and control tweets. We illustrate the variation in relative frequencies in Fig 2, a wordshift plot that demonstrates the most influential words (by frequency and happiness) driving the happiness benefit [24]. As mentioned above, positive words (with a happiness score greater than 6) including *beautiful, fun, enjoying*, and *amazing* appeared more frequently in in-parks tweets.

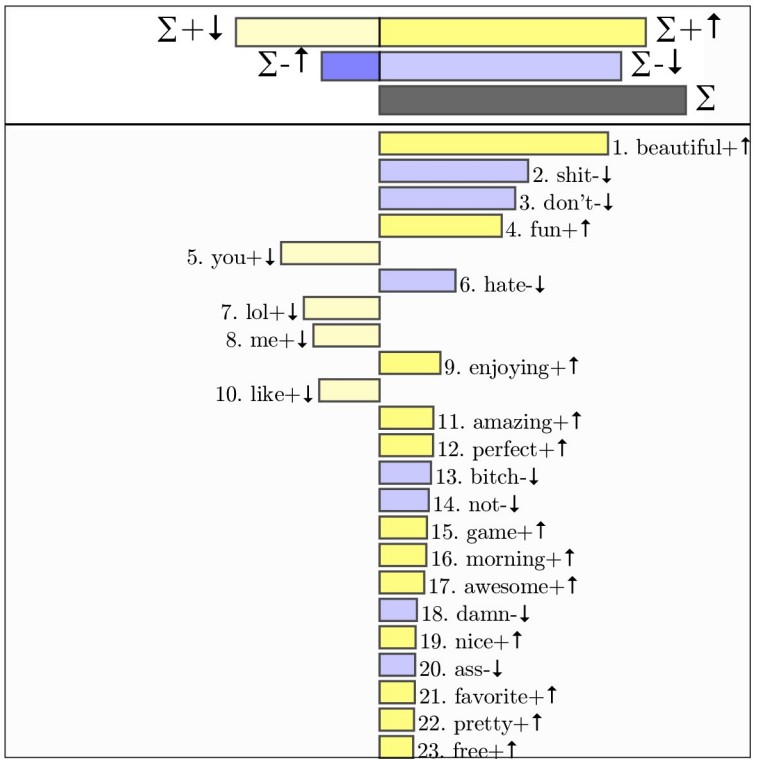

**Fig 2. Wordshift diagram between park and control tweets.** Differences in word frequency between park and control tweets across all cities, in order of decreasing contribution to the difference in average happiness. The right side represents the park tweets, with an average happiness of 5.96. The left side represents the control tweets, with an average happiness of 5.86. Purple bars represent words ≤ 4 (with − symbol) on the Hedonometer scale. Yellow bars represent words ≥ 6 (with + symbol) on the Hedonometer scale. Arrows indicate whether a word was more or less frequent within that set of tweets compared to the other text. For example, *beautiful* is a positive word (yellow) with higher frequency in in-park tweets that contributes to its higher average happiness than the control tweets. *Don't* is a negative word (purple) that appears less frequently in in-park tweets, also resulting in a higher average happiness score compared to control groups. Going against the overall trend, the positive words *lol* and *me* are used less often in parks. This wordshift uses tweets from 1,000 random in-park tweets and 1,000 control tweets from each city.

Negative words (with a happiness score less than 4) such as *don't, not* and *hate* appeared less frequently in in-park tweets. Interactive versions of individual city wordshift graphs are available in the online appendix accompanying this manuscript at http://compstorylab.org/cityparkhappiness/.

## Park analysis

We plot the mean happiness benefit values against two metrics of park quality—park spending and ParkScore® (Fig 3). There is no clear pattern between happiness benefit and park spending or ParkScore®, contrary to what we hypothesized in **H2A and H2B**. Interestingly, Indianapolis, which had the highest mean happiness benefit, had the lowest municipal park spending per capita and one of the lowest ParkScore® values. Washington D.C., San Francisco, Chicago, New York, and Seattle had the highest ParkScore® values, and were all fairly close to the mean happiness benefit of 0.10.

Park spending per capita and ParkScore® were not correlated with mean happiness benefit by city. However, prior work has demonstrated an association between park investment and levels of self-rated health [20]. Another study found higher levels of physical activity and health

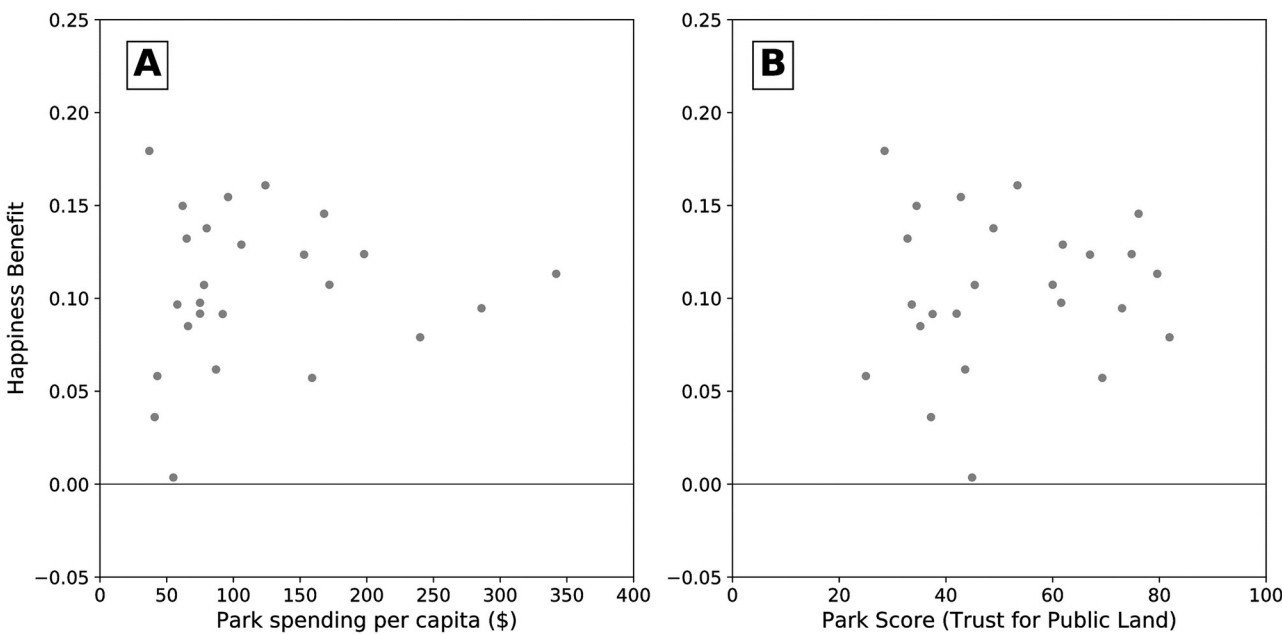

**Fig 3. Park analysis and happiness benefit. A**. Park spending per capita vs mean happiness benefit by city. Park spending per capita is from Trust for Public Land (TPL) data. **B**. ParkScore® vs happiness benefit. The TPL calculates ParkScore® annually from measures of park acreage, access, investment, and amenities, and is scaled to a maximum score of 100. The happiness benefit was not strongly correlated with per capita spending (Spearman's $\rho = 0.14$) or ParkScore® (Spearman's $\rho = 0.03$).

to be associated with a composite score of park quality in 59 cities [34]. Other factors such as heterogeneous use patterns of Twitter across cities may be more associated with happiness benefit than measures of park quality and spending. We call for further investigation into the relationship between park quality and investment with the mental health benefits of nature contact.

We grouped in-park tweets into four categories based on the size of the park and estimated the happiness benefit for each category to test **H3** (Fig 4A). Parks greater than 100 acres had

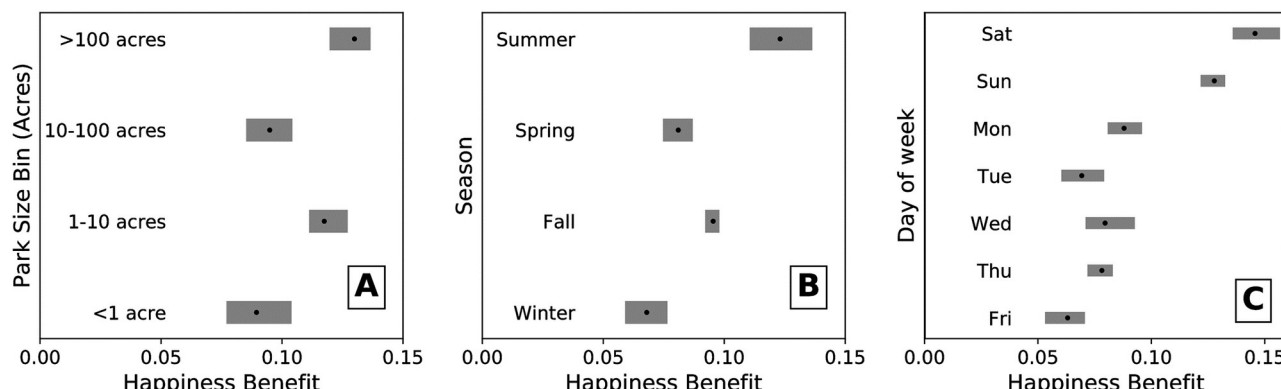

**Fig 4. Temporal analysis of happiness benefit. A**. Happiness benefit by park size. The largest category of parks (greater than 100 acres) had the highest happiness benefit. **B**. Happiness benefit by season, with summer and fall exhibiting the highest mean happiness benefit values. **C**. Happiness benefit by day of the week, with the weekend days higher than other days of the week. In all three panels, the range is the full range of happiness benefits from 10 runs, sampling 80% of tweets. 1,000 random in-park tweets were pooled in each group from each city. Control tweets were selected as tweets most temporally proximate to the in-park tweet from the same city.

the highest mean happiness benefit of 0.13, followed by parks from 1 − 10 acres (0.12). Parks less than 1 acre and parks between 10 − 100 acres had the lowest mean happiness benefit of 0.09.

Tweets inside of all park size categories exhibited a positive happiness benefit. The largest parks, greater than 100 acres, had the highest mean happiness benefit. One possible explanation is that larger parks provide greater opportunities for mental restoration and separation from the taxing environment of the city. This finding is consistent with results from our earlier study in San Francisco, in which tweets in the larger and greener Regional Parks had the highest happiness benefit [18]. Parks between 0 and 10 acres are often neighborhood parks that people use in their day to day lives. Local parks provide many essential functions; however, our results suggest that the experiences people have in larger parks may be more beneficial from a mental health perspective. Another possibility is that people spend more time in larger parks; one study suggested that 120 minutes of nature contact a week resulted in improved health and well-being [35].

## Temporal analysis

Across all cities, we grouped park tweets and their control tweets according to the in-park tweet's timestamp to test **H4**. First, we compared the happiness benefit by season. The mean happiness benefit was highest in the summer (0.12), followed by fall (0.10), spring (0.08), and winter (0.06) as shown in Fig 4B. Then we grouped park tweets and their respective control tweets according to the day of the week in which it was posted. Saturday exhibited the highest mean happiness benefit (.15) followed by Sunday (0.13). Monday through Friday were all between 0.06 and 0.09 (Fig 4). We also estimated the happiness benefit by hour of the day. The tweets posted during the 8:00 and 9:00 AM hours had a mean happiness benefit around 0.07 while the rest of the day did not show a clear pattern, ranging from 0.08 to 0.14 (S4 Fig).

We observe that the mean happiness benefit was higher in summer than other seasons; however, the happiness benefit was positive in all four seasons. Possible interpretations of seasonal differences may include that warmer or sunnier weather in the summer leads to an increased benefit from park visitation. People may engage in longer visits to parks during summer months, engage in physical activity, or connect with friends during the summer, all of which may increase the benefits of spending time in a park [36]. Alternatively, more non-residents may be tweeting from parks during the summer, leading to greater within-park sentiment scores. Similar dynamics may be driving the higher happiness benefits on the weekend compared to weekdays, though all days of the week exhibited positive values (See Fig 4). Prior work has shown that people on Twitter are happiest on the weekends and during times of year with more daylight [37]. Nevertheless, our comparisons indicate that a sentiment benefit occurs throughout the day, week, and year, indicating that the effect is not purely driven by temporal patterns. Our hourly comparison indicates that a sentiment benefit occurs during all hours of the day, indicating that the effect is not purely driven by leaving the office. This result is encouraging because some prior studies on nature contact using Twitter analyzed shorter time periods. Future studies should seek methods that can investigate the other temporal aspects of nature contact including the frequency and duration of visits [38].

We acknowledge that studying human behavior using Twitter data involves several potential sources of bias. Active users on Twitter tend to be younger and more affluent than the population at large [39]. Instead of investigating how individual users and demographic subgroups respond to nature contact, we attempt to estimate the aggregate effect of park visitation on happiness across a city. While our happiness benefit calculation uses same-city tweets as a

control, the results may not generalize beyond Twitter users. We only use English language tweets which may limit our ability to generalize to other languages and cultures. We do not control for nearby demographics when assessing the happiness benefit of specific parks. For example, larger parks may be promixal to more affluent neighborhoods or associated with adjacent neighborhood age structure. While this may introduce bias across parks within cities, it should not impact our results comparing the total happiness benefit across cities.

## Future directions

Our results, along with those from previous studies, point to several important areas of future research. Future research should continue to explore the relationship between tweet happiness and other factors beyond park investment. While ParkScore® captures a variety of park-quality related metrics, vegetation and biodiversity are salient features of greenspace that significantly impact how people experience their time in nature [40–42].

More localized studies could look at the mental health impact of park-level vegetative cover and biodiversity metrics. Alternatively, similar methods could be applied to compare the mental benefits of nature contact with other experiences such as museum visits or sports games. This could provide insight into the benefit of investing in public goods such as parks for health outcomes relative to alternatives. Similarly, these analyses could isolate the importance of experiencing nature compared to the social and cultural factors that influence sentiment on Twitter.

While we investigated the seasonal variation of in-park happiness, climate and weather have been shown to influence happiness on Twitter as well [43, 44]. Tweets could be binned by some composite of temperature, humidity, and precipitation in order to investigate how weather moderates the association between nature contact and mental well-being [21].

Demographic, socioeconomic, and cultural factors also play a role in how people engage with parks [45]. While identifying such factors on Twitter is challenging and requires ethical consideration, other methodologies can continue to explore how different groups use and benefit from time in parks, to help ensure that the benefits of parks are available to everyone. As the evidence continues to mount on the many different benefits of nature contact, we must ensure park access to quality parks for all urban residents.

## Supporting information

**S1 Appendix. Gauging the happiness benefit of US urban parks through Twitter.** (ZIP)

**S1 Fig. Normalized histogram of LabMT words and stop words taken out of the analysis due to being in a park name.** Our analysis is conservative as the ratio is higher for positive words ($> 6$) compared to negative words ($< 4$). Words between 4 and 6 are not included in our analysis. (TIF)

**S2 Fig. Happiness benefit by city.** Happiness benefit by city. We derive each city's full range of values from 10 bootstrap runs, for which we randomly selected 80% of tweets. Darker dots represent mean value from bootstrap runs. For each city, the control group consists of 1 random, non-park tweet from each user paired with an in-park tweet. (TIF)

**S3 Fig. User control plots.** A. The left panel shows park spending per capita vs mean happiness benefit by city. Park spending per capita is from Trust for Public Land (TPL) data. B. The right panel shows ParkScore® vs mean happiness. The TPL calculates ParkScore® annually

from measures of park acreage, access, investment, and amenities, and is scaled to a maximum score of 100.
(TIF)

**S4 Fig. Change in happiness benefit by hour of day.** The range is the full range of happiness benefit estimates from 10 runs, sampling 80% of tweets. 1,000 random in-park tweets were pooled in each group from each city. Control tweets were selected as tweets most temporally proximate to the in-park tweet from the same city.
(TIF)

## Author Contributions

**Conceptualization:** Aaron J. Schwartz, Peter Sheridan Dodds, Taylor H. Ricketts, Christopher M. Danforth.

**Data curation:** Aaron J. Schwartz, Jarlath P. M. O'Neil-Dunne, Christopher M. Danforth.

**Formal analysis:** Aaron J. Schwartz.

**Investigation:** Aaron J. Schwartz.

**Methodology:** Aaron J. Schwartz, Jarlath P. M. O'Neil-Dunne, Christopher M. Danforth.

**Resources:** Aaron J. Schwartz.

**Supervision:** Peter Sheridan Dodds, Taylor H. Ricketts, Christopher M. Danforth.

**Visualization:** Aaron J. Schwartz, Peter Sheridan Dodds.

**Writing – original draft:** Aaron J. Schwartz.

**Writing – review & editing:** Aaron J. Schwartz, Peter Sheridan Dodds, Jarlath P. M. O'Neil-Dunne, Taylor H. Ricketts, Christopher M. Danforth.

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
