## [Decision Letter · Decision Letter 0]

26 Nov 2020

PONE-D-20-27777

Gauging the happiness benefit of US urban parks through Twitter

PLOS ONE

Dear Dr. Schwartz,

Thank you for submitting your manuscript to PLOS ONE. After careful consideration, we feel that it has merit but does not fully meet PLOS ONE’s publication criteria as it currently stands. Therefore, we invite you to submit a revised version of the manuscript that addresses the points raised during the review process.

We look forward to receiving your revised manuscript.

Kind regards,

Mingxing Chen, Ph.D.

Academic Editor

PLOS ONE

Journal Requirements:

"We are grateful for support from the NSF GRFP program, the Gund Institute for the Environment Catalyst Award program, and a gift from MassMutual Life Insurance Company."

"AJS is supported by NSF GRFP program DGE1451866. https://www.nsfgrfp.org/

All authors are supported by Gund Institute for the Environment Catalyst Award program. https://www.uvm.edu/gund/gund-catalyst-awards

Additionally, because some of your funding information pertains to commercial funding, we ask you to provide an updated Competing Interests statement, declaring all sources of commercial funding.

In your Competing Interests statement, please confirm that your commercial funding does not alter your adherence to PLOS ONE Editorial policies and criteria by including the following statement: "This does not alter our adherence to PLOS ONE policies on sharing data and materials.” as detailed online in our guide for authors  http://journals.plos.org/plosone/s/competing-interests.  If this statement is not true and your adherence to PLOS policies on sharing data and materials is altered, please explain how.

Please include the updated Competing Interests Statement and Funding Statement in your cover letter. We will change the online submission form on your behalf.

4. We note that Figure 4 in your submission contains map images which may be copyrighted. All PLOS content is published under the Creative Commons Attribution License (CC BY 4.0), which means that the manuscript, images, and Supporting Information files will be freely available online, and any third party is permitted to access, download, copy, distribute, and use these materials in any way, even commercially, with proper attribution. For these reasons, we cannot publish previously copyrighted maps or satellite images created using proprietary data, such as Google software (Google Maps, Street View, and Earth). For more information, see our copyright guidelines: http://journals.plos.org/plosone/s/licenses-and-copyright.

(1) You may seek permission from the original copyright holder of Figure 4 to publish the content specifically under the CC BY 4.0 license. 

Reviewers' comments:

Reviewer's Responses to Questions

**Comments to the Author**

1. Is the manuscript technically sound, and do the data support the conclusions?

Reviewer #1: Yes

Reviewer #2: Partly

Reviewer #3: Yes

2. Has the statistical analysis been performed appropriately and rigorously? 

Reviewer #1: Yes

Reviewer #2: N/A

Reviewer #3: Yes

3. Have the authors made all data underlying the findings in their manuscript fully available?

Reviewer #1: Yes

Reviewer #2: No

Reviewer #3: Yes

4. Is the manuscript presented in an intelligible fashion and written in standard English?

Reviewer #1: Yes

Reviewer #2: Yes

Reviewer #3: Yes

5. Review Comments to the Author

Reviewer #1: The analysis in this article is very innovative and meaningful, it combines the Twitter platform and big data technology with the relationship between language and emotion, to explore the impact of urban parks and happiness benefits. However, several aspects need to be further explored.

1. The content of the literature review is not sufficient enough, and the paper does not accurately point out what previous studies have been done, and which aspects need to be filled in.

2. The titles of the figures are too long, adding the explanations of the figures in the text or adding some legends to the figure is more appropriate.

3. More methods can be considered in the article to determine the relationship between green space and happiness in different geographical locations, such as geographically weighted regression.

Reviewer #2: This study estimated a happiness benefit, the difference in expressed happiness between in- and out-of-park tweets, for 25 cities in the US. The topic of this paper is worthwhile to be explored. However, the analysis is too simple to make the conclusions. （1）This manuscript lacks necessary theoretical thinking.（2）The literature review is too simple. There is little introduction and comment on the results of the analysis. (3) The whole article is very descriptive, without much solid and in-depth analysis.

Reviewer #3: The paper is an interesting analysis of social media data. I suggest several minor revisions:

page 2: 'These pathways have been explored using a dose-response framework which describe the duration, frequency, and intensity of nature contact.' should be cited.

page 2 etc: Please call out the hypotheses more directly, using bold text or H1: xxx, so the reader is able to quickly access these important statements.

page 3, The first full paragraph seems to support another hypothesis.

page 4: Sentiment analysis - provide a sentence that explains the derivation of word scores as this information is fundamental to the paper's premise. Sentence at end of same paragraph is unclear.

Concluding Remarks: You may wish to address these observations and questions:

- This is another study providing correlational connections between nature and health. Could there be a more refined methodology using social media that would reveal causal mechanisms?

- As I read I found myself thinking, what would be the happiness treets scores for people in a museum, or at an event in a sports stadium, or even with friends at a bar? What would be happiness quotients comparisons? And considering that there would be similarities what would be the cost/benefit ratio of investing in different facilities to promote happiness?

6. PLOS authors have the option to publish the peer review history of their article (what does this mean?). If published, this will include your full peer review and any attached files.

Reviewer #1: No

Reviewer #2: No

Reviewer #3: **Yes: **Kathleen L Wolf

---

## [Author Response · Author response to Decision Letter 0]

21 May 2021

Editors

PLOS ONE

May 21, 2021

Dear Dr. Chen:

We thank you and the reviewers for the constructive feedback and for the opportunity to revise and resubmit. We have addressed all comments and are pleased to submit our revised manuscript, Gauging the happiness benefit of US urban parks through Twitter, for inclusion in the Urban Ecosystems collection at PLOS ONE. We have attached our responses to both editor and reviewer comments below. 

We have submitted a fully reformatted manuscript as well as a marked-up version produced with latexdiff showing revisions from the original submission. 

We appreciate your consideration and look forward to your response.

Best regards,

Aaron J. Schwartz

Aaron.J.Schwartz@colorado.edu

Peter Dodds

Jarlath O’Neil-Dunne

Taylor Ricketts

Chris Danforth

 

We have adapted PLOS ONE’s style requirements. We have included a version of the original manuscript submission (old_manuscript.pdf) in PLOS ONE format, the revised manuscript in PLOS ONE format manuscript.pdf) and a marked-up version of the manuscript produced with latexdiff (revised_manuscript_with_track_changes.pdf).

"We are grateful for support from the NSF GRFP program, the Gund Institute for the Environment Catalyst Award program, and a gift from MassMutual Life Insurance Company."

"AJS is supported by NSF GRFP program DGE1451866. https://www.nsfgrfp.org/

All authors are supported by Gund Institute for the Environment Catalyst Award program. https://www.uvm.edu/gund/gund-catalyst-awards

Additionally, because some of your funding information pertains to commercial funding, we ask you to provide an updated Competing Interests statement, declaring all sources of commercial funding.

Here is an updated Funding Statement: 

AJS is supported by NSF GRFP program DGE1451866. https://www.nsfgrfp.org/

All authors are supported by Gund Institute for the Environment Catalyst Award

program. https://www.uvm.edu/gund/gund-catalyst-awards

Authors associated with the MassMutual Center for Excellence for Complex Systems and Data Science are supported in part by a gift from MassMutual Life Insurance Company.

The funders had no role in study design, data collection and analysis, decision to

publish, or preparation of the manuscript.

In your Competing Interests statement, please confirm that your commercial funding does not alter your adherence to PLOS ONE Editorial policies and criteria by including the following statement: "This does not alter our adherence to PLOS ONE policies on sharing data and materials.” as detailed online in our guide for authors http://journals.plos.org/plosone/s/competing-interests. If this statement is not true and your adherence to PLOS policies on sharing data and materials is altered, please explain how.

Please include the updated Competing Interests Statement and Funding Statement in your cover letter. We will change the online submission form on your behalf.

Updated Competing Interests Statement:

Authors associated with the MassMutual Center for Excellence for Complex Systems and Data Science are supported in part by a gift from MassMutual Life Insurance Company. This does not alter our adherence to PLOS ONE policies on sharing data and materials.

Data will be made available upon acceptance at the following DOI: 10.6084/m9.figshare.12915329

4. We note that Figure 4 in your submission contains map images which may be copyrighted. All PLOS content is published under the Creative Commons Attribution License (CC BY 4.0), which means that the manuscript, images, and Supporting Information files will be freely available online, and any third party is permitted to access, download, copy, distribute, and use these materials in any way, even commercially, with proper attribution. For these reasons, we cannot publish previously copyrighted maps or satellite images created using proprietary data, such as Google software (Google Maps, Street View, and Earth). For more information, see our copyright guidelines: http://journals.plos.org/plosone/s/licenses-and-copyright. 

(1) You may seek permission from the original copyright holder of Figure 4 to publish the content specifically under the CC BY 4.0 license. 

We have decided to remove the figure at this time to avoid copyright issues. 

 

Response to reviewers:

Reviewer #1: The analysis in this article is very innovative and meaningful, it combines the Twitter platform and big data technology with the relationship between language and emotion, to explore the impact of urban parks and happiness benefits. However, several aspects need to be further explored.

1. The content of the literature review is not sufficient enough, and the paper does not accurately point out what previous studies have been done, and which aspects need to be filled in.

Response: We have expanded our literature review and reorganized our introduction section (Lines 2-91). We now discuss relevant literature from four different types of approaches to studying nature contact (experimental, epidemiological, experience-based, social media) and explain the contributions of each to the field. We have more clearly explained what prior studies have done when examining urban parks and happiness, as well as the most relevant gaps. 

2. The titles of the figures are too long, adding the explanations of the figures in the text or adding some legends to the figure is more appropriate.

Response: We have shortened the figure titles (See Figures 1-5). We have maintained detailed captions to ensure readers can fully parse the figures.

3. More methods can be considered in the article to determine the relationship between green space and happiness in different geographical locations, such as geographically weighted regression.

Our sample of 25 cities does not lend itself to regression methods. We are not trying to estimate a causal relationship between greenspace and happiness across locations. Instead, we have performed a variety of controls on our sentiment analysis – matching Tweets by time of day within the same city and user (Materials & Methods, S2. Appendix), removing a variety of bots and hashtags (S3. Appendix) and performing the bootstrap procedure in our estimate of happiness benefit. These methods help us compare and explore the relative mental benefit of urban park visitation across cities.

Reviewer #2: This study estimated a happiness benefit, the difference in expressed happiness between in- and out-of-park tweets, for 25 cities in the US. The topic of this paper is worthwhile to be explored. However, the analysis is too simple to make the conclusions. 

(1) This manuscript lacks necessary theoretical thinking. 

Thank you for your suggestions. We have added theoretical grounding to the introduction (Lines 19-47)

(2）The literature review is too simple. 

We have significantly broadened the literature review, organized into 4 paragraphs (Lines 9-47)

(3) There is little introduction and comment on the results of the analysis. 

We integrated our results and discussion sections into a single section, which we introduce on lines 186-189. We comment on the results of the analysis throughout this section (Lines 197-286). 

Reviewer #3: The paper is an interesting analysis of social media data. I suggest several minor revisions:

page 2: 'These pathways have been explored using a dose-response framework which describe the duration, frequency, and intensity of nature contact.' should be cited.

Thank you for catching this omission. During our revision of the introduction and theoretical framework we removed this sentence. The original citation should have been:

Shanahan, D. F., Fuller, R. A., Bush, R., Lin, B. B., & Gaston, K. J. (2015). The Health Benefits of Urban Nature: How Much Do We Need? BioScience, 65(5), 476–485. https://doi.org/10.1093/biosci/biv032

page 2 etc: Please call out the hypotheses more directly, using bold text or H1: xxx, so the reader is able to quickly access these important statements.

Thank you, we have added 5 explicit hypotheses bolded and marked as H1-H4 on lines 55,60, 65, 69, 79. We now reference these in our results as well (see lines 192, 238-239, 252, 270).

page 3, The first full paragraph seems to support another hypothesis.

We added an additional explicit hypothesis on line 60 (H2B).

page 4: Sentiment analysis - provide a sentence that explains the derivation of word scores as this information is fundamental to the paper's premise. Sentence at end of same paragraph is unclear.

Thank you for this comment. We have edited the paragraph on Lines 119-133 to provide additional detail on the derivation of word scores and clarified the last sentence of the paragraph.

Concluding Remarks: You may wish to address these observations and questions:

This is another study providing correlational connections between nature and health. Could there be a more refined methodology using social media that would reveal causal mechanisms?

This is a key challenge in non-experimental approaches to studying nature contact and well-being. There are many determinants of mental health and it may be impossible to completely isolate the contribution of nature to someone’s well-being. We added additional detail to our introduction on the pros and cons of different methodological approaches (Lines 20-47) to clarify the utility of social media-based approaches and how they can contribute to collective efforts in studying nature and health. We attempt to address some of the correlational issues by including control groups for our in-park tweets, which we discuss in Materials & Methods (Lines 134-145) and expand upon in the Appendix (Lines 465-478).

As I read I found myself thinking, what would be the happiness tweet scores for people in a museum, or at an event in a sports stadium, or even with friends at a bar? What would be happiness quotients comparisons? And considering that there would be similarities what would be the cost/benefit ratio of investing in different facilities to promote happiness?

Thank you, this is a great suggestion. We now include a call for future research of this type in our Future Directions section (Lines 294-300). We note that there would be interesting methodological challenges to overcome such as data density (museums) or in filtering out stop words (sporting events) that should be informed by domain specific knowledge.

---

## [Decision Letter · Decision Letter 1]

24 Aug 2021

PONE-D-20-27777R1

Gauging the happiness benefit of US urban parks through Twitter

PLOS ONE

Dear Dr. Schwartz,

Thank you for submitting your manuscript to PLOS ONE. After careful consideration, we feel that it has merit but does not fully meet PLOS ONE’s publication criteria as it currently stands. Therefore, we invite you to submit a revised version of the manuscript that addresses the points raised during the review process.

We look forward to receiving your revised manuscript.

Kind regards,

Mingxing Chen, Ph.D.

Academic Editor

PLOS ONE

Journal Requirements:

Reviewers' comments:

Reviewer's Responses to Questions

**Comments to the Author**

1. If the authors have adequately addressed your comments raised in a previous round of review and you feel that this manuscript is now acceptable for publication, you may indicate that here to bypass the “Comments to the Author” section, enter your conflict of interest statement in the “Confidential to Editor” section, and submit your "Accept" recommendation.

Reviewer #4: (No Response)

2. Is the manuscript technically sound, and do the data support the conclusions?

Reviewer #4: (No Response)

3. Has the statistical analysis been performed appropriately and rigorously? 

Reviewer #4: (No Response)

4. Have the authors made all data underlying the findings in their manuscript fully available?

Reviewer #4: (No Response)

5. Is the manuscript presented in an intelligible fashion and written in standard English?

Reviewer #4: (No Response)

6. Review Comments to the Author

Reviewer #4: Based on the sentiment trend analysis from Twitter data, this paper studies the happiness scores in and outside parks in 25 cities in the United States, then calculates the happiness benefits. It’s a carefully done study and the results are of considerable interest, but there are still some issues to be clarified as follows.

(1) This paper uses the weighted average method to calculate the happiness benefits, which often conceals the high and low values in the areas outside the park. Is it possible to use thermograms to show the spatial distribution of annual happiness within these cities?

(2) In the chapter of Temporal Analysis, the discussion of related results was missed. For example, the results in summer are higher than winter, and in weekends are higher than working days. It is worth considering that is this related to people’s travel habits.

(3) We are happy to see your great results, but there is a lack of discussion at the end of this paper about the limitations, which is a common part of big data research. For example, the data comes from younger people who are more likely to share Twitter, and whether the higher spatial scale of the park (Fig4. A) is related to the surrounding population density and age structure is worth further discussion.

7. PLOS authors have the option to publish the peer review history of their article (what does this mean?). If published, this will include your full peer review and any attached files.

Reviewer #4: No

---

## [Author Response · Author response to Decision Letter 1]

27 Sep 2021

Editors

PLOS ONE

September 27, 2021

Dear Dr. Chen:

We thank you and the reviewers for the constructive feedback and for the opportunity to revise and resubmit. We have addressed all comments and are pleased to submit our revised manuscript, Gauging the happiness benefit of US urban parks through Twitter, for inclusion in the Urban Ecosystems collection at PLOS ONE. We have attached our responses to comments from Reviewer #4 below. In the prior revision, we made significant changes based on feedback from Reviewers #1-3.

We have submitted a fully reformatted manuscript as well as a marked-up version produced with latexdiff showing revisions from the previous submission. 

We appreciate your consideration and look forward to your response.

Best regards,

Aaron J. Schwartz

Aaron.J.Schwartz@colorado.edu

Peter Dodds

Jarlath O’Neil-Dunne

Taylor Ricketts

Chris Danforth

 

Reviewer #4: Based on the sentiment trend analysis from Twitter data, this paper studies the happiness scores in and outside parks in 25 cities in the United States, then calculates the happiness benefits. It’s a carefully done study and the results are of considerable interest, but there are still some issues to be clarified as follows.

(1) This paper uses the weighted average method to calculate the happiness benefits, which often conceals the high and low values in the areas outside the park. Is it possible to use thermograms to show the spatial distribution of annual happiness within these cities?

Thank you for this thoughtful comment. While the weighted average may conceal high and low values, our method of analysis purposefully does this to ask a specific question about the relative effect of park visitation on happiness as compared to other urban environments. While producing location-specific thermograms is beyond the scope of this paper, we have acknowledged the potential obscuring of extreme values in the manuscript on lines 210-212.

“Prior work has shown that tweet happiness can vary within a city (even down to the 

neighborhood level) and extreme sentiment values may be obscured by our weighted averaging procedure \\cite{gibbons2019twitter}. We chose an aggregated approach to detect an overall signal about the effect of parks on happiness, but understanding detailed spatial patterns in happiness is an important future research direction.”

(2) In the chapter of Temporal Analysis, the discussion of related results was missed. For example, the results in summer are higher than winter, and in weekends are higher than working days. It is worth considering that is this related to people’s travel habits.

This is a helpful suggestion. We have added several sentence and an additional citation further expanding the discussion of our temporal analysis on lines 285-297.

“Possible interpretations of seasonal differences may include that warmer or sunnier weather in the summer leads to an increased benefit from park visitation. People may engage in longer visits to parks during summer months, engage in physical activity, or connect with friends during the summer, all of which may increase the benefits of spending time in a park \\cite{Pretty2017}. Alternatively, more non-residents may be tweeting from parks during the summer, leading to greater within-park sentiment scores. Similar dynamics may be driving the higher happiness benefits on the weekend compared to weekdays, though all days of the week exhibited positive values (See Fig.~\\ref{fig_bins}). Prior work has shown that people on Twitter are happiest on the weekends and during times of year with more daylight \\cite{golder2011diurnal}. Nevertheless, our comparisons indicate that a sentiment benefit occurs throughout the day, week, and year, indicating that the effect is not purely driven by temporal patterns.”

(3) We are happy to see your great results, but there is a lack of discussion at the end of this paper about the limitations, which is a common part of big data research. For example, the data comes from younger people who are more likely to share Twitter, and whether the higher spatial scale of the park (Fig4. A) is related to the surrounding population density and age structure is worth further discussion.

Thank you, we have expanded our discussion and added a paragraph on the limitations of Twitter data around demographic representation. We have also included text discussing the limitations of our specifc methodology. For instance, we acknowledge that there are some confounding factors and potential biases in our approach of pooling tweets around specific types of parks. Please find these additions on lines 303-314.

“We acknowledge that studying human behavior using Twitter data involves several potential sources of bias. Active users on Twitter tend to be younger and more affluent than the population at large \\cite{blank2017digital}. Instead of investigating how individual users and demographic sub-groups respond to nature contact, we attempt to estimate the aggregate effect of park visitation on happiness across a city. While our happiness benefit calculation uses same-city tweets as a control, the results may not generalize beyond Twitter users. We only use English language tweets which may limit our ability to generalize to other languages and cultures. We do not control for nearby demographics when assessing the happiness benefit of specific parks. For example, larger parks may be promixal to more affluent neighborhoods or associated with adjacent neighborhood age structure. While this may introduce bias across parks within cities, it should not impact our results comparing the total happiness benefit across cities.”

---

## [Decision Letter · Decision Letter 2]

24 Nov 2021

Gauging the happiness benefit of US urban parks through Twitter

PONE-D-20-27777R2

Dear Dr. Schwartz,

We’re pleased to inform you that your manuscript has been judged scientifically suitable for publication and will be formally accepted for publication once it meets all outstanding technical requirements.

Kind regards,

Mingxing Chen, Ph.D.

Academic Editor

PLOS ONE

Additional Editor Comments (optional):

Reviewers' comments:

Reviewer's Responses to Questions

**Comments to the Author**

1. If the authors have adequately addressed your comments raised in a previous round of review and you feel that this manuscript is now acceptable for publication, you may indicate that here to bypass the “Comments to the Author” section, enter your conflict of interest statement in the “Confidential to Editor” section, and submit your "Accept" recommendation.

Reviewer #4: All comments have been addressed

2. Is the manuscript technically sound, and do the data support the conclusions?

Reviewer #4: Partly

3. Has the statistical analysis been performed appropriately and rigorously? 

Reviewer #4: Yes

4. Have the authors made all data underlying the findings in their manuscript fully available?

Reviewer #4: (No Response)

5. Is the manuscript presented in an intelligible fashion and written in standard English?

Reviewer #4: Yes

6. Review Comments to the Author

Reviewer #4: The author made a good reply to the comments on the review of the manuscript, and made corresponding changes to the content of the article one by one, adding to the limitations of the existing data and methods. In spite of this, it is still a good article to explore the direction of emotional perception in urban space.

7. PLOS authors have the option to publish the peer review history of their article (what does this mean?). If published, this will include your full peer review and any attached files.

Reviewer #4: No

---

## [Editor Report · Acceptance letter]

3 Mar 2022

PONE-D-20-27777R2 

Gauging the happiness benefit of US urban parks through Twitter 

Dear Dr. Schwartz:

I'm pleased to inform you that your manuscript has been deemed suitable for publication in PLOS ONE. Congratulations! Your manuscript is now with our production department. 

Kind regards, 

on behalf of

Prof. Mingxing Chen 

Academic Editor

PLOS ONE